# Biomass-Derived Carbons as Versatile Materials for Energy-Related Applications: Capacitive Properties vs. Oxygen Reduction Reaction Catalysis

**Stefan Breitenbach [1,2,*], Nemanja Gavrilov [3], Igor Pašti [3], Christoph Unterweger [1], Jiri Duchoslav [4], David Stifter [4], Achim Walter Hassel [2] and Christian Fürst [1]**

[1]  Wood K plus-Kompetenzzentrum Holz GmbH, 4040 Linz, Austria; c.unterweger@wood-kplus.at (C.U.); c.fuerst@wood-kplus.at (C.F.)

[2]  Institute of Chemical Technology of Inorganic Materials (TIM), Johannes Kepler University Linz, 4040 Linz, Austria; achimwalter.hassel@jku.at

[3]  Faculty of Physical Chemistry, University of Belgrade-Faculty of Physical Chemistry, Studentski trg 12-16, 11158 Belgrade, Serbia; gavrilov@ffh.bg.ac.rs (N.G.); igor@ffh.bg.ac.rs (I.P.)

[4]  Center for Surface and Nanoanalytics (ZONA), Johannes Kepler University Linz, 4040 Linz, Austria; jiri.duchoslav@jku.at (J.D.); david.stifter@jku.at (D.S.)

[*]  Correspondence: s.breitenbach@wood-kplus.at; Tel.: +43-732-2468-6785

**Abstract:** Biomass-derived carbons are very attractive materials due to the possibility of tuning their properties for different energy-related applications. Various pore sizes, conductivities and the inherent presence of heteroatoms make them attractive for different electrochemical reactions, including the implementation of electrochemical capacitors or fuel cell electrodes. This contribution demonstrates how different biomass-derived carbons prepared from the same precursor of viscose fibers can reach appreciable capacitances (up to 200 F g$^{-1}$) or a high selectivity for the oxygen reduction reaction (ORR). We find that a highly specific surface area and a large mesopore volume dominate the capacitive response in both aqueous and non-aqueous electrolytic solutions. While the oxygen reduction reaction activity is not dominated by the same factors at low ORR overpotentials, these take the dominant role over surface chemistry at high ORR overpotentials. Due to the high selectivity of the O$_2$ reduction to peroxide and the appreciable specific capacitances, it is suggested that activated carbon fibers derived from viscose fibers are an attractive and versatile material for electrochemical energy conversion applications.

**Keywords:** biomass-derived carbons; capacitance; energy conversion; oxygen reduction reaction

## 1. Introduction

The emerging energy requirements of our society, combined with the predicted exhaustion of fossil fuel reserves, necessitates the development of efficient energy conversion and storage systems. Electrochemical power sources are at the front of this battle, providing different energy and power density ranges for various mobile and stationary applications [1].

The core of any electrochemical power source is the electrode material, and different classes of materials are used in electrochemical capacitors, batteries, and fuel cells. Carbon materials find application in all three types of electrochemical power sources, but have different roles. In electrochemical capacitors, carbons are, as a rule, the active materials, while in batteries and fuel cells, they serve predominantly as conductive additives or catalyst supports [2–4]. The family of carbon materials is vast and exceptionally rich, especially at the nanoscale where nanocarbons exist in the 1D [5,6], 2D [7,8] or 3D form [9–11]. However, while carbon materials can be made at almost atomic precision using sophisticated synthetic routes, such materials are rather expensive and unsuitable for large-scale production.

Biomass-derived carbons are produced via the conversion of biomass using chemical or thermal treatments [12–14]. During this conversion, the naturally occurring hetero

elements from the biomass are frequently incorporated into the carbon structures, leading to rich surface chemistry suitable for different electrochemical applications. Moreover, some heteroatoms can be introduced into the structure by intentionally adding the source of the given elements so that the desired properties, such as the chemical composition, can be reached [15–17]. In addition, the pore size distribution and the specific surface area can be effectively tuned using different chemical impregnation agents and temperature treatments or using post-synthetic activation [15,18]. In this way, the physical and chemical properties of biomass-derived carbons can be effectively controlled.

However, considering the practical application of carbons in electrochemical energy conversion, it is still unclear which properties of the carbon materials rule their overall performance [19,20]. The debate in the scientific community is very active and is typically directed towards one of the many possible applications. For example, in terms of capacitive applications, the most relevant factors influencing the response are: morphology, pore structure, specific surface area, heteroatom doping, degree of graphitization, and the presence of defects [21]. However, rather different, and sometimes opposing, claims can be found in the literature regarding the role of pore size and structure, the presence and the type of the heteroatoms (usually nitrogen, sulfur, phosphorus. and boron) [22], specific surface area, and other parameters [23]. Moreover, it is not yet clear how the joint effects of the different properties of these materials affect the overall performance for a given application. However, considering the wide applicability of carbons [24,25] and the attractive properties of biomass-derived carbons [26], such as the cost and the possibility to upscale the synthesis, we believe that different applications of carbon materials should be considered jointly.

In this contribution, we first demonstrate the synthesis of carbons derived from viscose fibers. Using different impregnation agents and activation conditions, materials with diverse chemical compositions and a wide range of specific surfaces were prepared. Materials were tested as capacitor electrodes and oxygen reduction reaction (ORR) catalysts. We show that the specific surface area, combined with a large fraction of mesopores, has the dominant role in the capacitive response and the ORR activity of these carbon materials. However, this only holds for high ORR overpotentials in the latter case, where the charge transfer is much faster than the mass transfer, while the specific surface is not decisive for the ORR onset potential.

## 2. Materials and Methods

### 2.1. Preparation of the ACFs

Viscose fibers (1.7 dtex, 38 mm, Lenzing AG, Lenzing, AUT) were dried overnight at 80 °C and then impregnated in a solution of either 4% diammonium hydrogen phosphate (DAHP) or 12% ammonium sulfate (AS) in deionized water for 15 min. Sørensen formol titration [27] was used to determine the fiber ammonium loading. Dried fibers (drying at RT) were carbonized in a chamber furnace under a nitrogen atmosphere at $T_{\text{carb}}$ = 850 °C. The heating rate was $v_{\text{carb}}$ = 10.0 °C min$^{-1}$ and an isothermal holding step of 20 min was applied after the final temperature was reached. The carbonized fibers were activated in a rotary kiln at $T_{\text{act}}$ = 850 °C for $T_{\text{act}}$ = 150–300 min in a CO$_2$ flow of $Q_{\text{CO2}}$ = 22.5 dm$^3$ h$^{-1}$. An overview of the samples and their preparation routes can be found in Table 1. Finally, the prepared activated carbon fibers (ACFs) were ground using a mortar mill (RM 200, Retsch GmbH, Haan, GER).

**Table 1.** Conditions of the sample preparation.

| Sample | Impregnation Agent | Loading | $T_{\text{carb}}$/°C | $v_{\text{carb}}$/°C min$^{-1}$ | $T_{\text{act}}$/°C | $t_{\text{act}}$/min | $Q_{\text{CO2}}$/dm$^3$ h$^{-1}$ |
|--------|-------------------|---------|------|------|------|------|------|
| P11 | 4% DAHP | 0.61% P | 850 | 10.0 | - | - | - |
| S12 | 12% AS | 2.33% S | 850 | 10.0 | - | - | - |
| P21 | 4% DAHP | 0.61% P | 850 | 10.0 | 850 | 150 | 22.5 |
| S22 | 12% AS | 2.33% S | 850 | 10.0 | 850 | 150 | 22.5 |
| S23 | 12% AS | 2.33% S | 850 | 10.0 | 850 | 300 | 22.5 |

### 2.2. Materials Characterization

The morphology of the samples was investigated using Scanning Electron Microscopy with Phenom ProX (Thermo Fisher Scientific, Waltham, MA, USA).

To determine the specific surface area and the textural structures, the obtained ACFs were analyzed by $N_2$ isothermal adsorption/desorption (77 K) on a gas sorption system (Autosorb iQ, Quantachrome Instruments, Boynton Beach, FL, USA). Before the analysis, samples were degassed for at least 2 h at 200 °C. The specific surface area and derived pore size distribution (PSD) were calculated using the Brunauer–Emmett–Teller (BET) method and the non-local density functional theory (NLDFT).

X-ray photoelectron spectroscopy (XPS) measurements were completed using a Theta Probe XPS system (Thermo Fisher Scientific, Waltham, MA, USA) controlled by the Avantage software package. The Avantage software was also used for data evaluation and the chemical assessment of the recorded spectra. The XPS system is equipped with a dual flood gun (FG02, Thermo Fisher Scientific, Waltham, MA, USA) providing a beam of low energy electrons (2 eV) and low energy Ar-ions simultaneously to suppress the negative effects caused by surface charging. A monochromated Al-K$\alpha$ X-ray source (1486.6 eV) was used and a spot size of 400 mm in diameter was used for the X-ray beam on the sample surface. The hemispherical analyzer of the system was operated in the CAE mode (constant analyzer energy) with a pass energy of 200 eV for the survey spectra and 20 eV for the high resolution (HR) spectra. For data evaluation, a Shirley-type background was subtracted from the photoelectron peaks. Elemental concentrations were determined from the peak areas, taking into account the instrument transmission function at the different pass energies, the relative sensitivity factors of the elements (based on calculations by Scofield [28]), and the different inelastic mean free paths of electrons at different kinetic energies (based on the TPP-2M method [28]). The HR spectra were fitted using a Gaussian–Lorentzian product function with a mixing parameter of 0.3. The full width at half maximum (FWHM) parameter was constrained to an interval between 1 and 2 eV.

Electrical resistivity measurements were performed across the fabricated electrodes (see Section 2.3) using a 4-point probe method. A cylindrical measuring head (SDKR-13, Nagy Instruments, Gäufelden, GER) holding 4 Rhodium-plated steel needles with a tip radius of 0.25 mm and in-line spacing of 1.25 mm was used. A sourcemeter (Keithley 2410, Tektronix GmbH, Cologne, GER) and a multimeter (Keithley 2750 Solon, Tektronix GmbH, Cologne, GER) were used for sourcing the current and measuring the potential drop, respectively. The applied current was varied between 0 and 3 mA in 0.1 mA steps. Then, the resistivity was calculated from the linear *I–V* dependence.

### 2.3. Electrode Preparation for Electrochemical Measurements

For electrochemical measurements in aqueous electrolytes, the ACFs were first ground in a mortar mill. Then, 5.0 mg of the selected carbon sample was suspended in 1 cm$^3$ of 40 *v/v*% ethanol/water solution and homogenized for 15 min in an ultrasonic bath. The prepared catalytic ink was drop-casted onto the glassy carbon (GC) disk electrode (cross-section surface 0.196 cm$^2$) and dried under $N_2$ flow. The overall loading of the ACFs was 250 µg cm$^{-2}$ per geometric surface area. Upon drying, the surface of the thin film was covered with 10 µL of 0.05 wt.% Nafion in ethanol to ensure its integrity during the electrochemical testing. The solvent was removed by evaporation.

For electrochemical measurements in a non-aqueous solution, the electrodes were prepared as follows. The prepared ACFs were ground using a mortar mill. Then, 3 wt.% polytetrafluoroethylene (PTFE) and 5 wt.% carbon black were added and the mixture was ground again for 15 min to obtain a kneadable dough. This dough was rolled out using a sheet metal roller to a thickness of 90–100 µm and circular electrodes with a radius of 1 cm were punched out. The prepared electrodes were dried in a vacuum for 1 h at 100 °C before being assembled for the measurements.

## 2.4. Electrochemical Measurements

Cyclic voltammetry (CV) was used to investigate the electrocatalytic and capacitive properties of doped carbons in aqueous solutions. A conventional one compartment three-electrode electrochemical cell, with a graphite rod as a counter electrode [29] and a saturated calomel electrode (SCE) as a reference electrode, was used. The electrode potentials were then recalculated to the Standard Hydrogen Electrode (SHE) scale ($E_{SCE}$ = 0.241 V vs. SHE). The capacitive performance was assessed in 3 mol dm$^{-3}$ KOH solution and 2 mol dm$^{-3}$ $H_2SO_4$ solution. A gentle gas flow of $N_2$ was kept just beneath the electrolyte surface to remove dissolved oxygen during capacitive measurements.

CV was also used to assess the capacitive properties of the prepared ACFs in the non-aqueous solution. Measurements were completed using a Vertex.One-potentiostat (Ivium Technologies B.V., Eindhoven, NL) at room temperature. The CV curves of the samples were measured with a scan rate of 20 mV s$^{-1}$ in the voltage window 0.0–2.0 V. The test setup consisted of two ACF electrodes soaked with the electrolyte (1.8 M triethyl methyl ammonium tetrafluoroborate (TEMA BF$_4$) in propylene carbonate (PC). A separator foil was inserted between the electrodes to prevent short circuit. Two carbon-coated foils were used as current collectors.

Gravimetric capacitances (in F g$^{-1}$) were evaluated using the following equation:

$$C = \frac{\int I\, dV}{m\, \Delta V\, v} \tag{1}$$

where $I$ (A) is the current, $\Delta V$ (V) is the potential window, $v$ (mV s$^{-1}$) is the scan rate, and $m$ (g) is the total mass of the active material.

Electrocatalytic activity toward the ORR was investigated in oxygen saturated in 0.1 mol dm$^{-3}$ KOH aqueous solution using rotating disk electrode (RDE) voltammetry. Measurements were completed using a Vertex.One-potentiostat (Ivium Technologies B.V., NL) with a Pine rotator (Pine research USA). The ORR was tested in an $O_2$-satured solution (purity 99.9995 vol.%) at room temperature. The measured currents were normalized to the geometrical cross-section area of the supporting GC disk.

To analyze the selectivity of the ORR process, RDE curves were processed using the Koutecky–Levich (KL) analysis [30,31]. Considering that the measured current is influenced by both electron and mass transfer (which are consecutive processes), the current ($j(E)$) can be presented as:

$$\frac{1}{j(E)} = \frac{1}{j_k(E)} + \frac{1}{j_d(\omega)} = \frac{1}{j_k(E)} - \frac{1}{B(E)\omega^{\frac{1}{2}}} \tag{2}$$

where $j_k(E)$ is the kinetic current and $j_d(E)$ is the diffusion current. The latter depends on the electrode rotation rate, the diffusion coefficient, and the concentration of $O_2$ and contains the number of electrons exchanged per $O_2$ molecule ($n$). This number depends on the electrode potential, so the diffusion current is also a function of the electrode potential. If all the other constants are known, the KL analysis allows the determination of the kinetic current and $n$; the latter gives the information about the selectivity of $O_2$ reduction to peroxide or OH$^-$.

## 3. Results

### 3.1. Physical and Chemical Properties of the Prepared CFs

#### 3.1.1. Morphology

All the samples had the same fibrous morphology. The carbon fibers were approx. 10 μm in diameter and, upon grinding, their length was typically around 200 μm. However, some longer fibers, as well as smaller debris, were present in all the samples. The low magnification SEM images of the samples are presented in Figure 1. We note that the surface of the fibers looks very smooth, even at high magnifications, showing no particular features, in line with previous findings [15].

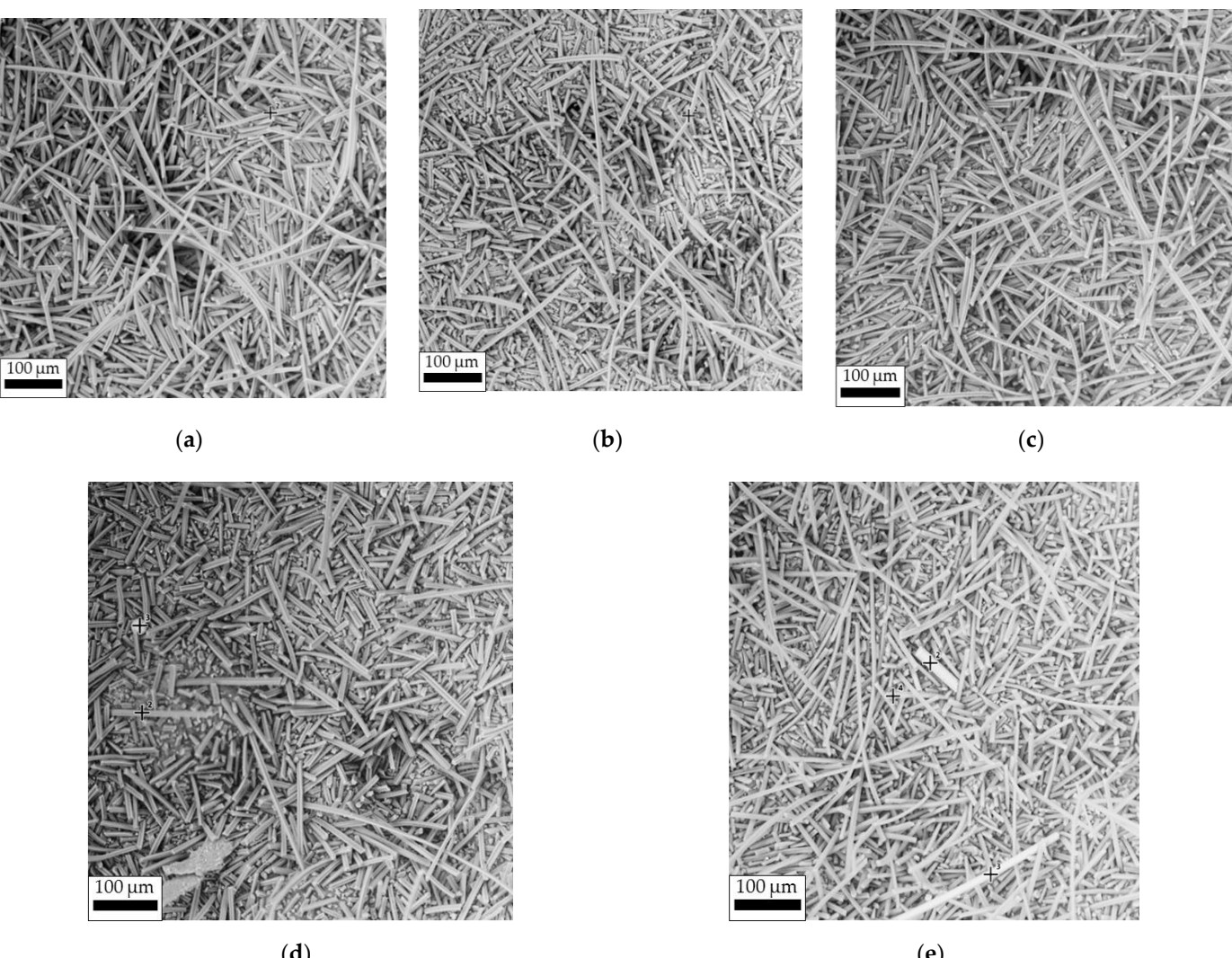

**Figure 1.** SEM images of the prepared CFs: (**a**) sample S12; (**b**) sample S22; (**c**) sample S23; (**d**) sample P11; (**e**) sample P21.

3.1.2. Textural Properties

The prepared porous carbon fibers produced from viscose were further investigated by nitrogen adsorption isotherms at 77 K. As shown in Figure 1, all samples displayed a sharp rise of the $N_2$ isotherm at the low-pressure range ($p/p_0 < 0.01$), which indicates the existence of a significant amount of micropores [32]. These observations were further confirmed by the PSD plot derived by NLDFT (Figure 2). The PSD of the carbonized fibers P11 and S12 show a similar pattern with a high pore volume < 1 nm diameter. Moreover, the specific surface area, the total pore volume, and the average pore width, shown in Table 2, are almost identical.

The PSD plots of the activated fibers, shown in Figure 2, show larger deviations. The DAHP-impregnated sample had a relatively large pore volume between 1 and 2.5 nm. The specific surface area of 2245 m$^2$ g$^{-1}$ is very high, as is the total pore volume of 0.88 cm$^3$ g$^{-1}$. The activated fibers, which were previously impregnated with AS, showed a lower pore volume. Doubling the activation time decreased the specific surface area from 667 m$^2$ g$^{-1}$ to 535 m$^2$ g$^{-1}$ and the total pore volume from 0.26 to 0.22 cm$^3$ g$^{-1}$. The average pore width (*d*) was also reduced from 0.75 nm to 0.50 nm.

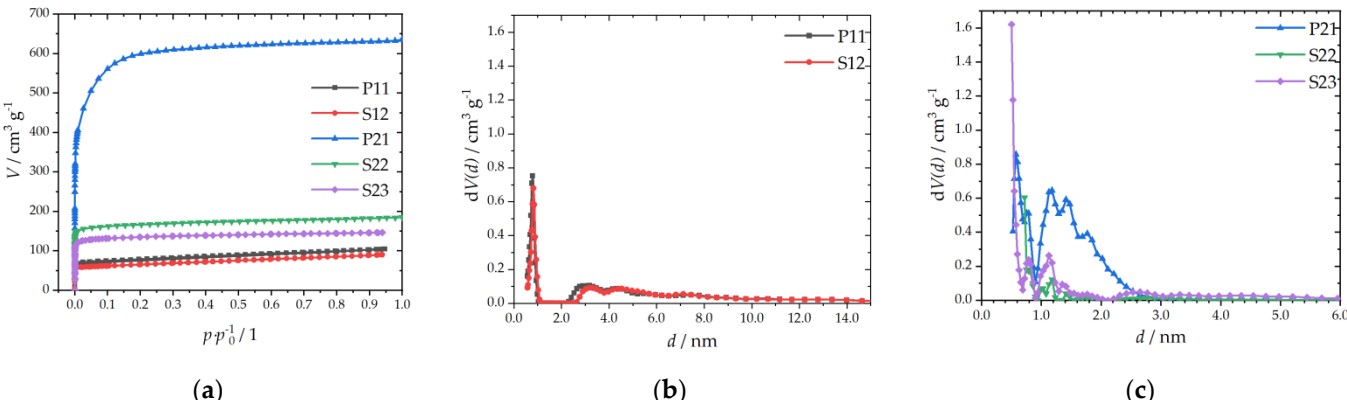

**Figure 2.** The results of the N$_2$ adsorption analysis: (**a**) Nitrogen adsorption isotherms of the ground fibers; (**b**) PSD of the carbonized fibers derived from the nitrogen adsorption isotherms by NLDFT; (**c**) PSD of the carbonized and activated fibers derived from the nitrogen adsorption isotherms by NLDFT.

**Table 2.** The results of the analysis of N$_2$ adsorption.

| Sample | $S_{BET}$ [a]/m$^2$ g$^{-1}$ | $V_{total}$ [b]/cm$^3$ g$^{-1}$ | $D_p$ [c]/nm |
|--------|------------------------------|----------------------------------|---------------|
| P11 | 292 | 0.15 | 0.79 |
| S12 | 252 | 0.13 | 0.82 |
| P21 | 2245 | 0.88 | 1.41 |
| S22 | 667 | 0.26 | 0.75 |
| S23 | 535 | 0.22 | 0.50 |

[a] Specific surface area estimated using the BET method; [b] Total pore volume calculated using the NLDFT method; [c] Average pore width calculated using the NLDFT method.

### 3.1.3. XPS Analysis

XPS was employed to investigate the elemental composition of the fibers (Table 3). DAHP treatment leads to detectable amounts of phosphorus (1 at.% to 2 at.%), even after activation. The higher phosphorus concentration in the ground fibers could point towards an increased phosphorus concentration in the fiber bulk. AS-impregnated fibers showed no sulfur but significant amounts of nitrogen (Figure 3a). As depicted in Figure 3b for sample S22 and summarized for all samples in Table 3, the chemical analysis of the C1s spectrum revealed the presence of aliphatic/aromatic CH$_3$/CH$_2$/CH such as carbons (peak at 285.0 eV), carbon bonded with nitrogen (286.0 eV), carbon bound in hydroxyl or/and ether groups (286.5 eV), C in the carbonyl group (287.9 eV), and C in the carboxyl group (289.5 eV). A weak peak found at 291.4 eV represents the π-π* shake-up satellite, which indicates the aromatic/graphitic character of the carbon material [33]. The N1s spectra of the nitrogen-containing samples showed nitrogen to be in two bonding motives of the pyridine (CN$_2$, peak at 398.8 eV) and graphitic (CN$_3$, 401.2 eV) ones (Table 3). The HR O1s spectra revealed two peaks–the one found at 533.4 eV is attributed to the C-O parts of hydroxyl, ether or/and carboxyl functionalities. The peak at the lower BE of 531.7 eV corresponds to the C=O parts of carboxyl and the carbonyl groups.

Doubling the activation time for AS-impregnated fibers slightly decreased the oxygen and nitrogen concentrations but had no visible effect on the carbon chemistry, as illustrated by the overlaid HR C1s spectra in Figure 3c.

**Table 3.** Elemental and chemical composition of the prepared CFs from the XPS analysis.

| Sample | C-C C-H | C-N | C | C-O | C=O | O-C=O | O | O-C O-P | O=C O=P | NC₂ | N | NC₃ | P | F | Na |
|---|---|---|---|---|---|---|---|---|---|---|---|---|---|---|---|
| | | | | | | | | | | | | | | | |

Elemental Concentration/at. % — Survey (Total)/HR

| Sample | C-C C-H | C-N | C | C-O | C=O | O-C=O | O | O-C O-P | O=C O=P | NC₂ | N | NC₃ | P | F | Na |
|---|---|---|---|---|---|---|---|---|---|---|---|---|---|---|---|
| P11 | 77.1 | - | 86.6 | 6.1 | 1.9 | 1.5 | 9.8 | 5.2 | 4.6 | - | - | - | 1.9 | 0.8 | 0.9 |
| S12 | 74.4 | 7.4 | 91.1 | 5.3 | 1.9 | 2.1 | 5.8 | 3.0 | 2.8 | 1.3 | 3.1 | 1.8 | - | - | - |
| P21 | 73.3 | 3.3 | 88.9 | 6.9 | 3.4 | 2.0 | 8.6 | 5.3 | 3.3 | 0.6 | 1.5 | 0.9 | 1.0 | - | - |
| S22 | 68.0 | 6.0 | 87.5 | 6.8 | 4.4 | 2.3 | 9 | 5.2 | 3.8 | 1.3 | 3.5 | 2.2 | - | - | - |
| S23 | 67.2 | 7.5 | 88.3 | 6.3 | 4.5 | 2.8 | 7.6 | 4.6 | 3.0 | 1.5 | 3.2 | 1.7 | - | 0.9 | - |

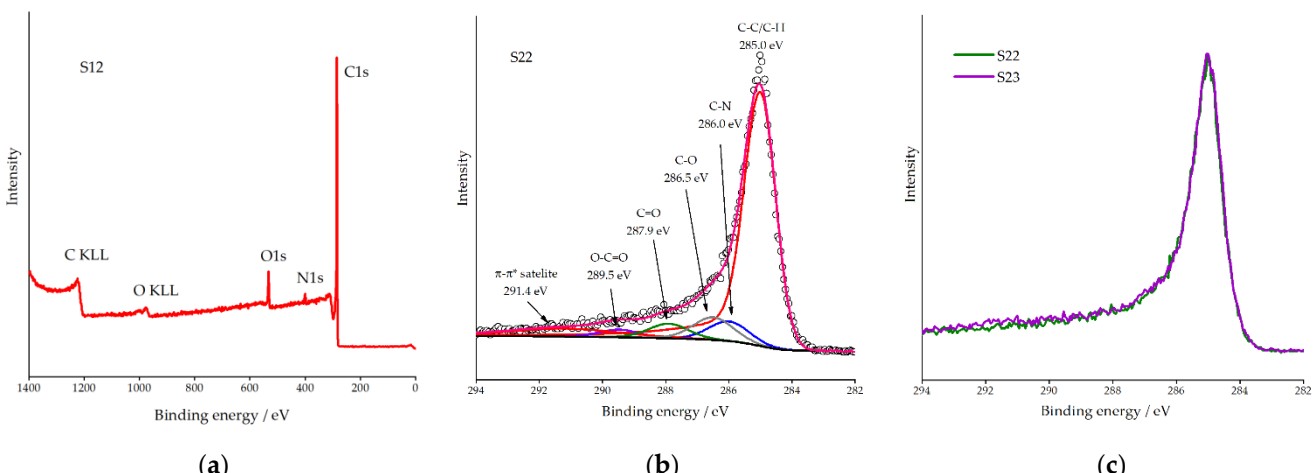

**Figure 3.** The results of the XPS analysis: (**a**) XPS survey spectrum of the ground sample S12; (**b**) chemical assessment of HR C1s spectrum of sample S22; (**c**) C 1s high resolution spectra of S22 and S23.

### 3.1.4. Electrical Resistivity

Electrical resistivity is crucial for good electrochemical performance as the material serves both as an active electrode material and a current collector; otherwise, significant energy losses and unnecessary self-heating can be expected if the inner resistance is too high. The results of the measurements of the electrical resistivity are shown in Table 4. The measurements were only carried out for the activated samples (P21, S22, and S23). Among these samples, the lowest resistivity was observed for the P21 sample, which contains P, in addition to C, O, and N. The lower resistivity for this sample could be due to the doping of the carbon network upon the incorporation of N and P atoms into the carbon structure [34,35] giving a combined effect and reducing resistivity in this way.

**Table 4.** Electrical resistivity ($\rho$) of the activated samples.

| Sample | $\rho/\Omega$ cm |
|---|---|
| P21 | 3.1 |
| S22 | 59.9 |
| S23 | 84.0 |

*3.2. Electrochemical Measurements*

Capacitive Properties

The capacitive properties were measured in an aqueous solution (KOH and $H_2SO_4$) using CV. In addition, a series of measurements were performed in a non-aqueous solution (TEMA $BF_4$ in PC).

Table 5 summarizes the measured gravimetric capacitances in aqueous solutions. In general, the capacitances measured in the KOH solution were larger than the ones measured in the $H_2SO_4$ solution. The CF samples which were not activated showed lower capacitances than the activated samples, particularly in the $H_2SO_4$ solution. The situation was similar in the KOH solution, although the differences were somewhat smaller. At first glance, there is a rough correlation between measured capacitance and the specific surface area of the samples. Non-activated samples had the smallest capacitance, followed by the activated samples with the intermediate specific surface area. The highest capacitances in both aqueous solutions were recorded for the P21 sample, which had the highest specific surface area among the studied samples (Table 2). Figure 4 presents the CV curves for the sample P21.

**Table 5.** Gravimetric capacitances of the investigated samples in aqueous solution at different potential scan rates (given in F g$^{-1}$).

| Scan Rate/mV s$^{-1}$ | KOH Solution | | | | | $H_2SO_4$ Solution | | | | |
|---|---|---|---|---|---|---|---|---|---|---|
| | S12 | S22 | S23 | P11 | P21 | S12 | S22 | S23 | P11 | P21 |
| 10 | 73 | 84 | 102 | 56 | 205 | 14 | 50 | 72 | 5.7 | 155 |
| 20 | 36 | 67 | 100 | 51 | 187 | 7.6 | 42 | 78 | 4.2 | 152 |
| 50 | 16 | 52 | 95 | 45 | 161 | 3.6 | 34 | 83 | 3.0 | 140 |
| 100 | 10 | 41 | 80 | 36 | 133 | 2.3 | 27 | 73 | 2.1 | 122 |
| 200 | 6.9 | 32 | 72 | 25 | 100 | 1.6 | 22 | 68 | 1.5 | 97 |

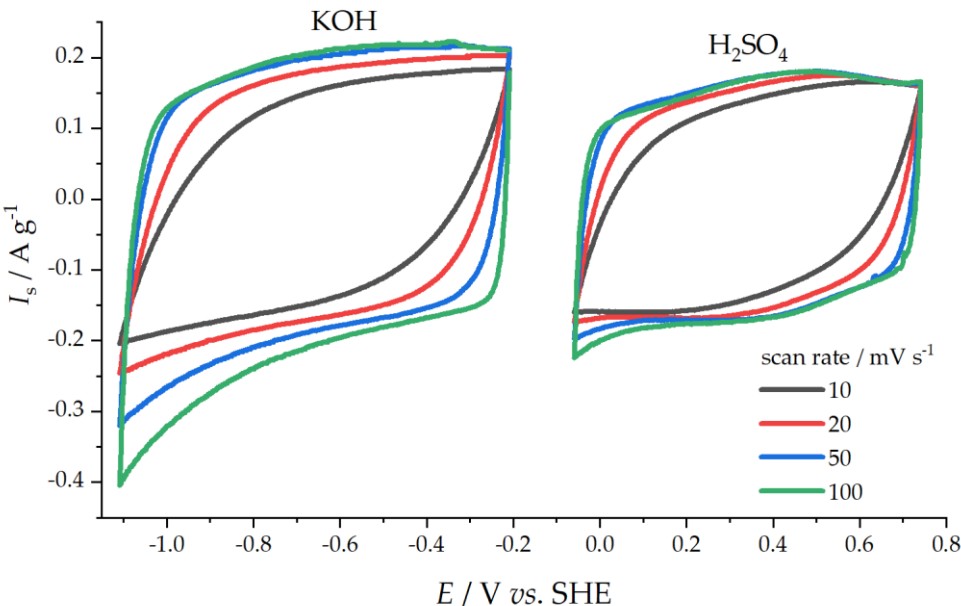

**Figure 4.** Cyclic voltammograms of P21 in KOH and $H_2SO_4$ solution at different potential scan rates.

Table 6 gives the measured capacitances of the investigated samples in a non-aqueous solution (TEMA $BF_4$ in PC), while Figure 5 compares the recorded cyclic voltammograms of P21 and P11. The electrodes made from the carbonized fibers P11 and S12 showed a low specific capacitance of 1.9 and 1.1 F g$^{-1}$, respectively. The specific capacitances of the activated AS-impregnated specimen were also low at 2–3 F g$^{-1}$. The electrode,

made of DAHP-impregnated activated fiber, had by far the highest specific capacitance of 127 F g$^{-1}$. The electrical conductivity of 3.1 $\Omega$ cm was significantly better than that of the other samples tested.

**Table 6.** Gravimetric capacitances of prepared CFs in 1.8 M TEMA BF4 in PC solution.

| Sample | *C*/F g$^{-1}$ |
| --- | --- |
| P11 | 1.9 |
| S12 | 1.1 |
| P21 | 127 |
| S22 | 2.1 |
| S23 | 2.9 |

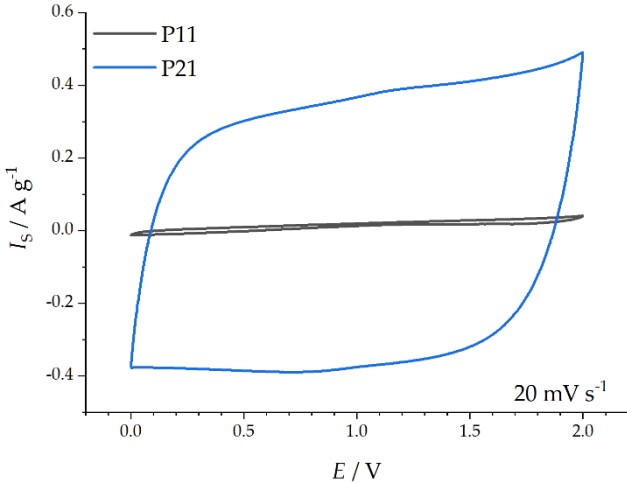

**Figure 5.** CV curves of samples P11 and P21 at the scan rate of 20 mV s$^{-1}$ in TEMA BF$_4$ in PC solution.

The recorded CV curves of the investigated CFs were processed using the method of Dunn [36] to evaluate the contributions of capacitive and pseudofaradaic processes in the overall capacitance. The results are presented in Table 7. The reported capacitive contributions were evaluated in the indicated potential windows (averaged from −0.759 to −0.559 V in alkaline and between 0.241 and 0.491 V vs. SHE in acidic solution) as due to the resistance of the studied materials, the fitting procedure was not accurate enough at potentials close to the anodic and cathodic vertex potentials. It can be seen that the capacitive component brings different contributions to the overall capacitance, and it is clear that surface chemistry contributes significantly through the pseudofaradaic process to the present surface functional groups.

**Table 7.** Capacitive current contribution in an alkaline solution averaged in the potential window from −0.759 to −0.559 V, and in acidic solution averaged between 0.241 and 0.491 V vs. SHE. The evaluation is completed for the sweep rate of 20 mV s$^{-1}$ and given in %.

| Sample | KOH Solution | H$_2$SO$_4$ Solution |
| --- | --- | --- |
| P11 | 99 ± 3 | 47.5 ± 5.7 |
| S12 | 52 ± 12 | 30 ± 10 |
| P21 | 58.3 ± 6.8 | 65.6 ± 8.3 |
| S22 | 17.8 ± 4.8 | 42.2 ± 3.2 |
| S23 | 51.9 ± 8.6 | 76.8 ± 3.0 |

*3.3. Oxygen Reduction Reaction Catalysis*

The ORR was investigated in an alkaline solution (0.1 mol dm$^{-3}$ KOH saturated with O$_2$). The obtained results show a large variety between the investigated samples in

terms of the ORR activity (Figure 6a). Taking the ORR onset potential ($E_{\text{onset}}$) as a measure of the catalytic activity (Table 8), the most active material was S23, with an ORR onset potential of −0.41 V vs. SHE (Figure 6b). This result is very important, as S23 had a much lower specific surface area than the P21 sample. It suggests that the ORR kinetics were not crucially determined by the surface area, at least at low ORR overpotentials close to the onset potential. Table 8 shows the evaluated kinetic currents and $n$ at different electrode potentials using Equation (2).

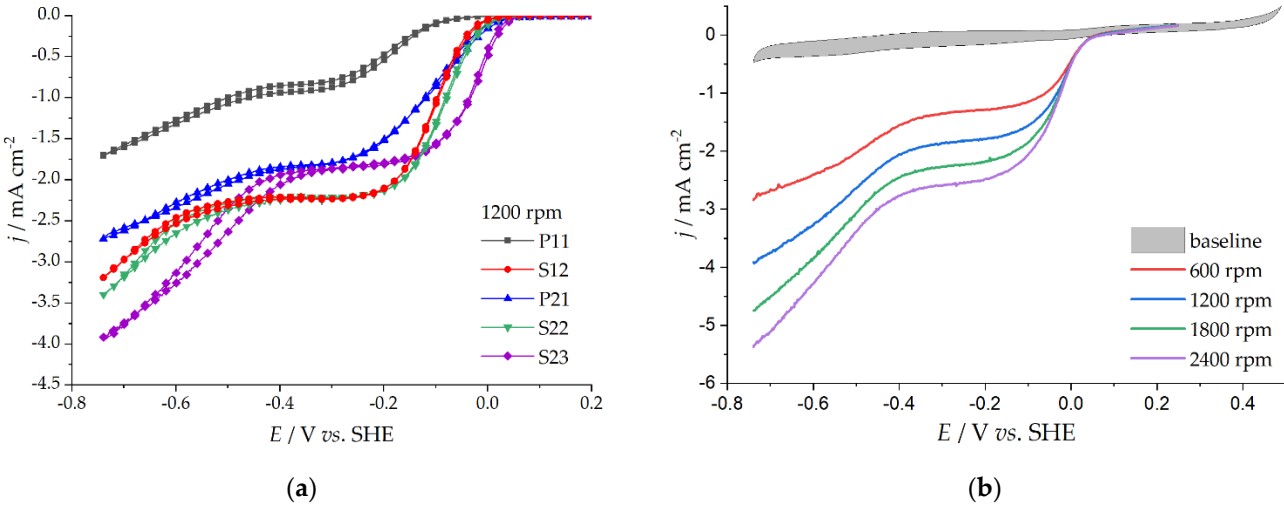

(a)           (b)

**Figure 6.** The results of the analysis of the ORR activity of studied samples in $O_2$-saturated 0.1 mol dm$^{-3}$ KOH solution: (**a**) background-corrected RDE-CV curves at a common electrode rotation rate of 1200 rpm for all the samples (potential scan rate 20 mV s$^{-1}$); (**b**) RDE-LSV curves of the S23 sample at different electrode rotation rates.

**Table 8.** Kinetic parameters for ORR for the studied samples: onset potential ($E_{\text{onset}}$), number of electrons ($n$) consumed per $O_2$ molecule, and kinetic currents ($j_k$).

|  |  | S21 | S22 | S23 | P11 | P21 |
|---|---|---|---|---|---|---|
| $E_{\text{onset}}$/V |  | −0.25 | −0.23 | −0.20 | −0.30 | −0.21 |
|  | $E$ / V vs. SHE |  |  |  |  |  |
| $n$ per $O_2$ molecule | −0.059 | 2.1 | 2.3 | 1.8 | - | - |
|  | −0.159 | 2.2 | 2.1 | 1.8 | - | 1.9 |
|  | −0.259 | 2.3 | 2.0 | 1.8 | 2.2 | 1.9 |
|  | −0.359 | 2.6 | 2.1 | 2.1 | 2.2 | 1.9 |
|  | −0.459 | 2.7 | 2.2 | 2.8 | 2.2 | 2.0 |
|  | −0.559 | 2.1 | 2.5 | 3.0 | 2.0 | 2.2 |
| $j_k$/ mA cm$^{-2}$ | −0.059 | 0.64 | 0.93 | 3.22 | 0.03 | 0.62 |
|  | −0.159 | 2.59 | 9.05 | 8.62 | 0.28 | 8.57 |
|  | −0.259 | 5.23 | 22.5 | 12.3 | 1.02 | 64.9 |

## 4. Discussion

To briefly sum up, the studied samples have identical fibrous morphologies but quite different specific surface areas. These results were achieved by using different impregnation agents and activation procedures. Moreover, using different impregnation agents, it was found that the chemical compositions were different. Nitrogen and phosphorus were effectively incorporated into the carbon structures, but we noted that the presence of sulfur was not confirmed using XPS. The highest specific surface area achieved was 2245 m$^2$ g$^{-1}$ for the sample impregnated with 4% DAHP and activated for 150 min, which was the sample denoted as P21.

When capacitive properties are of interest, we can see that the specific surface area played a dominant role among the studied samples. The sample P21 with the highest specific surface area had the largest specific capacitance in any of the studied electrolytes,

both aqueous and non-aqueous. In aqueous electrolytes, we can see that a significant contribution of the pseudofaradaic process can be seen. For example, the sample P21 in the KOH solution (Table 7) showed that ~58% of its capacitance comes from double-layer charging. Using the values of the measured capacitances and a usual value for areal capacitance (20 μF cm$^{-2}$), we can estimate that the surface of roughly 500 m$^2$ g$^{-1}$ was exposed to the electrolyte, so not the entire specific surface area. Overall, in the non-aqueous electrolyte, the measured BET surfaces correlate with the calculated specific capacitances. In addition, the samples P11, S12, S22, and S23 had the most pores with a pore diameter of less than 1 nm. However, this pore volume is not accessible to the organic electrolyte, which leads to a reduced specific capacitance [37]. The sample P21 had a large pore volume with a pore diameter of 1–2.5 nm. The resulting surface can be used to form Helmholtz double layers, thus achieving a comparatively high specific capacitance.

It is important to note that in an aqueous solution, the measured capacitances were larger than in non-aqueous solutions, which is particularly important for practical application. Namely, most of the studies of capacitive properties of different materials, including carbons, are performed in aqueous solutions. However, commercially available electrochemical capacitors use, as a rule, non-aqueous electrolytes because higher energy densities are achieved due to a wider potential window. Taking the values for P21 at 20 mV s$^{-1}$ (Tables 5 and 6), we estimated energy densities of 16.6 Wh kg$^{-1}$ (KOH solution), 17.1 Wh kg$^{-1}$ (H$_2$SO$_4$ solution), and 71 Wh kg$^{-1}$ (non-aqueous solution). At first glance, these values are quite appreciable when a typical range of energy densities of electrochemical capacitors is considered [38]. However, these values were evaluated using only the mass of the active material, which is more or less usual practice in scientific literature. These values would drop significantly for a realistic electrochemical device when the masses of current collectors, separator, and casing were considered.

Considering the ORR, its activity can be estimated using several approaches. First, one can take $E_{onset}$ as the measure of electrocatalytic ORR activity. If this parameter is used (Table 8), then it is clear that the specific surface is not the dominant factor for ORR as the sample S23 had the highest $E_{onset}$. The reported onset potentials were quite high, and the studied materials, except P11, stand side by side to other carbons reported in the literature [22,39–41].

Considering the observed ORR selectivity, the studied materials all favored 2 e$^-$ reduction of O$_2$ to peroxide, making them rather attractive for direct electrochemical peroxide generation [42]. However, another parameter, kinetic current, has to be taken into account as well. We can see that when the cathode polarization increased, the kinetic current rose, but at different rates. At −0.259 V vs. SHE (which is −0.5 V vs. SCE and 0.54 V vs. Reversible Hydrogen Electrode for pH = 13.5), the activated sample beat the non-activated ones and the kinetic current roughly scaled to the specific surface. Mass activities are easily obtained from kinetic currents by dividing the kinetic currents with the catalyst loading (Figure 7). We can see that the sample P21 topped out with a mass activity of ~250 A g$^{-1}$, followed by the samples S22 and S23. Hence, we conclude that upon reaching high cathodic polarization when the charge transfer rate is much higher than the mass transfer, the specific surface takes the dominant role over the surface chemistry and determines the overall ORR activity. For this reason, we believe that sample P21 would be a perfect candidate for the electrochemical generation of peroxide, where diffusion limitations are less pronounced compared to the RDE experiments.

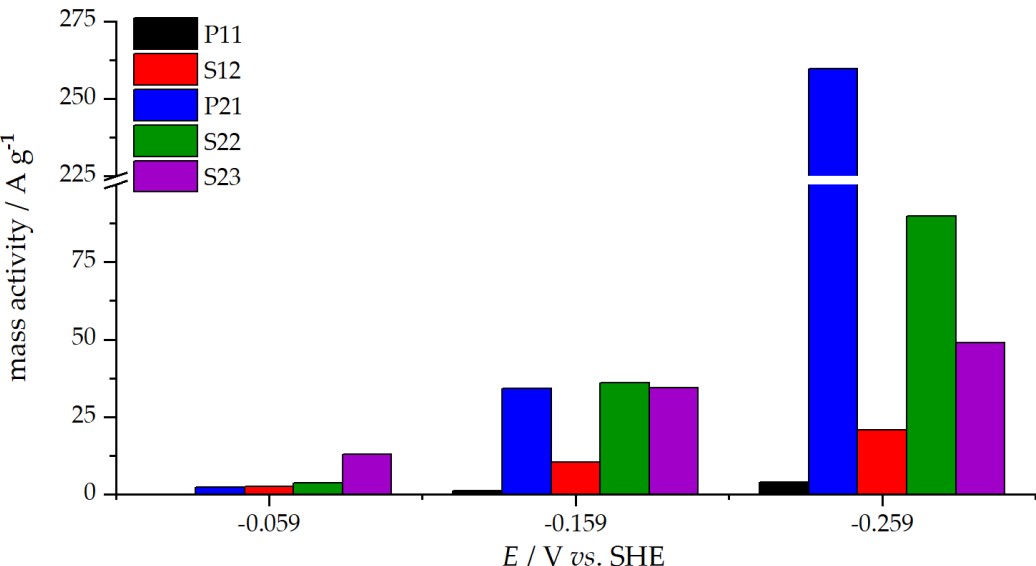

**Figure 7.** Mass-specific ORR activities of investigated samples at three different potentials.

## 5. Conclusions

Using viscose fibers as a precursor, five carbon materials were produced. By employing different impregnation agents and activation steps, the produced carbons resulted in different chemical compositions (incorporating nitrogen or phosphorus, besides carbon and oxygen) and specific surfaces ranging from 292 to 2245 $m^2$ $g^{-1}$. We found that the capacitive properties of the studied materials were dominated by the specific surface–the material with the largest specific surface and a large fraction of mesopores. In aqueous KOH electrolytic solutions, capacitance tops at 205 F $g^{-1}$, while in a $H_2SO_4$ solution capacitance tops at 155 F $g^{-1}$. Our analysis suggests that only a fraction of the surface is available to the electrolyte for this material, and we expect that this is the external surface and the surface of mesopores. Namely, in a non-aqueous solution, 1.8 M TEMA BF4 in PC, only the material with a large fraction of mesopores displayed a high capacitance of 127 F $g^{-1}$, while the dominant microporous materials had capacitances below 3 F $g^{-1}$. In addition, all the materials were active for the oxygen reduction reaction in an alkaline KOH solution, favoring 2 $e^-$ reduction to peroxide, and the specific surface area did not dominate the ORR onset potential. However, at higher ORR overpotentials, the specific surface became dominant, and the materials with higher specific surfaces displayed a higher mass activity for $O_2$ reduction to peroxide. Again, the highest mass activity was observed for the material with the largest fraction of mesopores. Hence, we suggest that versatile, biomass-derived carbons for energy conversion application can be effectively produced using strategies to enhance the mesopore contribution while incorporating heteroatoms, such as nitrogen and phosphorus, to improve the capacitive response through the pseudofaradaic processes and increased electrical conductivity.

**Author Contributions:** Conceptualization, S.B.; Methodology, S.B., C.U., D.S., J.D., A.W.H., N.G., I.P. and C.F.; Investigation, S.B., J.D. and N.G.; Resources, C.U., D.S. and A.W.H.; Data Curation, J.D., S.B. and N.G.; Writing—Original Draft Preparation, S.B. and I.P.; Writing—Review and Editing, C.U., D.S., A.W.H., N.G. and I.P.; Visualization, S.B. and I.P.; Supervision, D.S., A.W.H., I.P. and C.F.; Project Administration, D.S. and C.F.; Funding Acquisition, C.F. All authors have read and agreed to the published version of the manuscript.

**Funding:** The authors wish to thank the European Regional Development Fund (EFRE) and the province of Upper Austria for the financial support of this study through the program IWB 2014-2020 (project BioCarb-K). N.G. and I.P. acknowledge the support provided by the Serbian Ministry of Education, Science, and Technological Development (Contract number: 451-03-68/2020-14/200146).

**Institutional Review Board Statement:** Not applicable.

**Data Availability Statement:** The data presented in this study are available on request from the corresponding author.

**Conflicts of Interest:** The authors declare no conflict of interest. The funders had no role in the design of the study; in the collection, analyses, or interpretation of data; in the writing of the manuscript, or in the decision to publish the results.

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
