# Peer review of "Biomass-Derived Carbons as Versatile Materials for Energy-Related Applications: Capacitive Properties vs. Oxygen Reduction Reaction Catalysis"

_carbon, 2021_

Round 1

Reviewer 1 Report

This manuscript is concerned with “Biomass-derived carbons as versatile materials for energy-related applications”. There is plagiarism of 22%. Here, I summarize some of my concerns regarding the manuscript:

  1. The author mentioned as “We find that high specific surface area and large mesopore volume dominates the capacitive response in both aqueous and non-aqueous electrolytic solutions….” which is the main finding in this paper. Whereas, it is already well known by literature that the influencing factors are Pore structure and specific surface area; Heteroatoms doping; Graphitization degree; Defects; Morphology etc. What is the novelty of the outcomes. Could you please! explain by differentiating with others work? see SusMat. 2021;1:211–240.
  2. It is noticed that the experimental sections 2, 2.1, 2.2, 2.3, 2.4 is just copy and paste that should be rewritten to reduce the similarity.
  3. Author discussed the preparation of porous carbon fibers and presented the SEM images with a resolution of 100 μm which did not show the porosity of the material. It is recommended to use the high-resolution images that clearly indicate the porosity of the material.
  4. The authors used XPS technique for elemental analysis but did not explain the results well. X-ray photoelectron spectroscopy (XPS) is a surface characterization technique that can analyze a sample to a depth of 2 to 5 nanometers (nm). XPS reveals which chemical elements are present at the surface and the nature of the chemical bond that exists between these elements. It is recommended that the authors should explain the chemical bond interactions. If the author only wants to show the elemental composition, then ICP results could be used for overall average composition.
  5. The authors used raw file of the XPS results in Figure 3 which is not clear. The authors should draw spectra from the raw file.
  6. All figures should be revised because the axis, font size, bold letter, graph structure did not match with each other. The author should keep a standard figure style and follow the consecutive format.

The manuscript should be revised in detail with proper argue and use better images before acceptance.

Reviewer 2 Report

The article creates a full, logical whole. The authors correctly planned the order of the analyses carried out. Research methodology is clearly described, indicating the subsequent stages of the work. The results correspond with the methodology, presenting the results of research in a transparent way. The subject of the publication is very interesting but there are some weak points that require adjustment:

  1. Line 61, 112: the closing brace is missing
  2. Line 225-230: The description of the Gravimetric capacitances estimation should be moved to the methodical part
  3. Line 286-290: The description of the current (j(E)) assessment should be moved to the methodical part

Round 2

Reviewer 1 Report

The authors improved the manuscript following reviewers' comments. Therefore, I will accept its publication. However, it is suggested that the authors should draw XPS spectra from the raw file.